# PE-DyRA: Dynamic Rank Adaptation for Parameter-Efficient Fine-Tuning via Importance-Aware Pruning and Expansion

## Abstract

As large language models grow in scale, full-parameter fine-tuning for downstream tasks incurs substantial computational and storage costs. Low-Rank Adaptation (LoRA) provides a parameter-efficient paradigm for model adaptation, but its fixed-rank allocation cannot adapt to the heterogeneous importance of different layers or the evolving requirements across training stages, resulting in either redundancy or insufficient capacity. In this paper, we introduce Dynamic Rank Adaptation via Importance-Aware Pruning and Expansion (PE-DyRA), a novel framework that dynamically allocates ranks through importance score-based pruning and expansion. PE-DyRA introduces three key innovations: 1) A parameter importance evaluation measure based on gradient information and input activations to enable more stable ranking; 2) A bidirectional rank adjustment mechanism that dynamically prunes and expands ranks based on importance, enabling flexible allocation and improved parameter utilization; 3)The PE-DyRA framework can be used as a paradigm to achieve better results on benchmark methods such as DoRA, PiSSA, and QLoRA. Extensive experiments demonstrate the effectiveness of PE-DyRA, surpassing baseline methods. Furthermore, theoretical analysis demonstrates that PE-DyRA has better parameter efficiency.

## 1 Introduction

Large language models (LLMs) (Achiam et al., 2023; Dubey et al., 2024; Guo et al., 2025) have become the core infrastructure of natural language processing, advancing performance from general-purpose inference to domain-specific applications (Ziems et al., 2023; Brown et al., 2020). However, the training and full parameter tuning of these models require huge computing resources and storage overhead (Raffel et al., 2020). Therefore, parameter-efficient fine-tuning methods for large-scale pre-trained models have become a research hotspot (Liu et al., 2022).

Low-Rank Adaptation (LoRA)(Hu et al., 2022) reduces computational cost by decomposing model weights into trainable low-rank matrices. However, its fixed-rank allocation limits adaptability and parameter efficiency (Yang et al., 2024), as different layers contribute unequally to downstream tasks. Kalajdzievski (2023) showed that increasing the rank of LoRA with proper scaling can significantly improve performance. Yet higher ranks incur substantial memory overhead, which has motivated the development of dynamic rank adaptation methods. DyLoRA (Valipour et al., 2023) employs random truncation to enable flexible inference-time rank selection, while IncreLoRA (Zhang et al., 2023a) incrementally allocates more parameters to important modules.AdaLoRA (Zhang et al., 2023b) prunes ranks via importance-based masking. AutoLoRA(Zhang et al., 2024) automates rank selection via meta-learned pruning of redundant singular components. TriAdaptLoRA (Liang et al., 2025) proposes an adaptive rank-growth strategy governed by dynamic thresholds. While the aforementioned methods are effective, they face two key limitations. First, they are restricted to either pruning or expansion. As shown in Figure 1a, pruning alone achieves high utilization but few effective ranks, whereas expansion increases effective ranks but with low utilization. Second, existing approaches primarily rely on weight or gradient magnitudes to assess parameter importance, often neglecting input activations, which play a crucial role in neuron outputs.

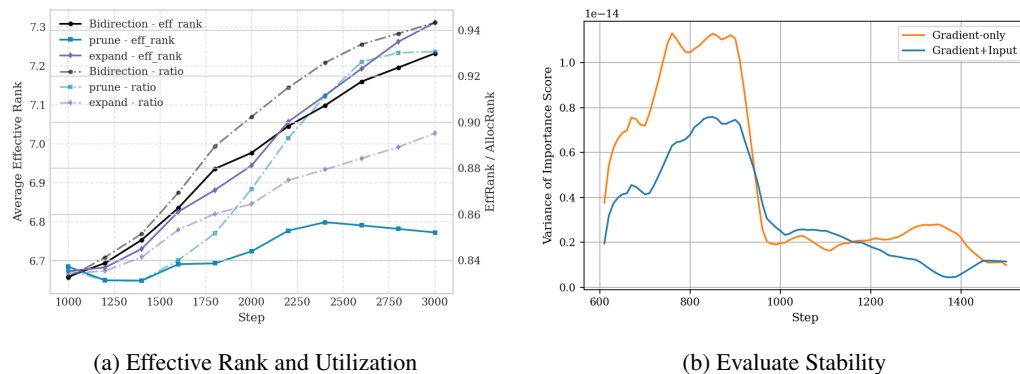

(a) Effective Rank and Utilization  (b) Evaluate Stability

Figure 1: Analysis of importance-guided dynamic rank allocation. (a) Effective rank and utilization ratio across training steps. (b) Stability evaluation of rank adaptation during training.

Table 1: Comparison of dynamic rank adaptation methods.

| Method | Importance Basis | Adaptation | Frequency |
|---|---|---|---|
| DyLoRA (Valipour et al., 2023) | Multi-rank joint training | No adjustment | Static |
| IncreLoRA (Zhang et al., 2023a) | Gradient | Expansion | Periodic |
| AdaLoRA (Zhang et al., 2023b) | Gradient | Pruning | Periodic |
| AutoLoRA (Zhang et al., 2024) | Meta-learning | Pruning | Post-optimization |
| TriAdaptLoRA (Liang et al., 2025) | Frobenius norms | Expansion | Periodic |
| **Ours** | **Gradient + Input** | **Bidirectional** | **Periodic** |

To address these limitations, we propose a novel dynamic rank assignment strategy that enables more efficient optimization of low-rank adapters through importance-based evaluation and adaptive rank adjustment. At scheduled intervals, parameter efficiency is improved by pruning redundant ranks and expanding those in critical layers. This strategy maintains both high effective rank and utilization during training(Figure 1a), and its bidirectional adjustment surpasses approaches restricted to pruning or expansion. For parameter importance evaluation, we combine gradient information, reflecting parameter sensitivity, with input activations, reflecting data dependence, to obtain a more accurate and fine-grained assessment. See Figure 1b, incorporating the input leads to a more stable evaluation throughout training. Table 1 provides a comparative overview of dynamic rank adaptation methods, demonstrating the advantages of our approach. We evaluate PE-DyRA across diverse tasks and model scales, consistently demonstrating superior performance over existing approaches.

The main contributions of this work are as follows:

1) A bidirectional rank adjustment mechanism that dynamically prunes and expands ranks based on importance, enabling flexible allocation and improved parameter utilization.

2) We propose an enhanced importance metric that integrates gradient-based parameter sensitivity with input activation information for more stable ranking.

3) PE-DyRA framework can be used as a paradigm on benchmark methods such as DoRA, PiSSA, and QLoRA to achieve better results.

4) Experimental results across diverse tasks demonstrate the effectiveness of PE-DyRA over baseline methods, while theoretical analysis confirms its superior parameter efficiency.

## 2 RELATED WORK

**Low-Rank Adaptation.** PEFT methods have evolved from adapter layers (Houlsby et al., 2019) and prompt tuning (Lester et al., 2021) to LoRA (Hu et al., 2022). The key insight of LoRA is that weight updates during adaptation can be effectively represented using low-rank decompositions.

PiSSA (Meng et al., 2024) leverages principal SVD to initialize LoRA by truncating pre-trained weights, thereby accelerating fine-tuning convergence. QLoRA (Dettmers et al., 2023) extends LoRA with 4-bit quantization and gradient dequantization, enabling efficient fine-tuning of large models. DoRA (Liu et al., 2024) decomposes the weight update into two independent components, amplitude and direction, and applies LoRA adaptation only to the direction component.

**Dynamic Rank Adaptation Methods.** Standard LoRA uses fixed-rank matrices, whereas recent work explores dynamic rank adaptation to optimize allocation during training. AdaLoRA (Zhang et al., 2023b) adapts ranks via a three-term SVD formulation with importance-based pruning and orthogonality regularization, but requires a large initial parameter space. DyLoRA (Valipour et al., 2023) enables flexible rank selection during inference by training a unified model across multiple candidate ranks simultaneously. IncreLoRA (Zhang et al., 2023a) instead adopts a progressive rank expansion strategy, gradually increasing the model capacity. SoRA (Ding et al., 2023) induces sparsity within LoRA modules. TriAdaptLoRA (Liang et al., 2025) draws on neuroscience principles to introduce an adaptive rank-growth strategy controlled by dynamic thresholds. These methods illustrate various strategies for dynamic LoRA rank allocation to balance efficiency and performance.

## 3 METHOD

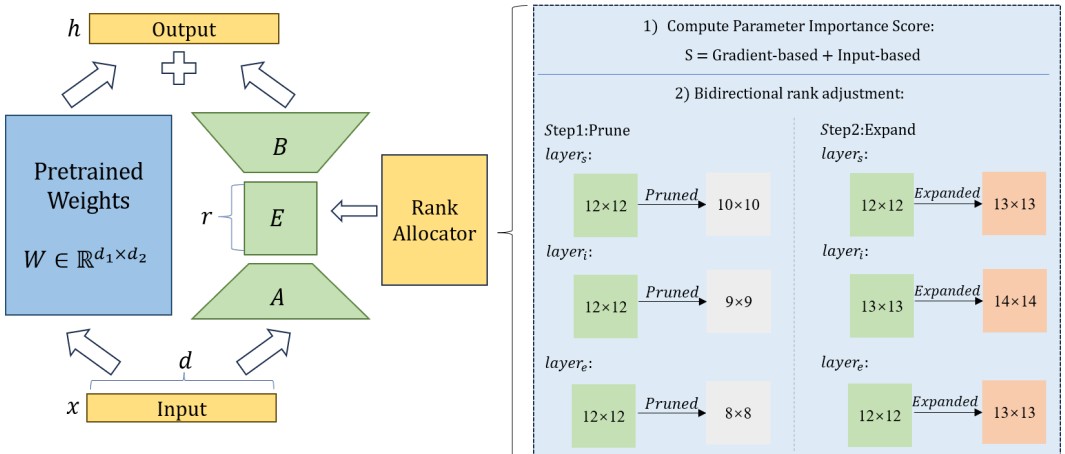

Figure 2: Overview of the proposed dynamic rank adjustment framework. (Left) The base architecture with rank allocation. (Right) The two-step bidirectional rank adjustment procedure: (1) compute parameter importance score $S$ based on both gradient and input information; (2) prune ranks in less important layers and expand them in more critical ones, enabling adaptive allocation of model capacity.

In this section, we propose PE-DyRA, a novel parameter-efficient fine-tuning method based on a dynamic rank adjustment framework that aims to dynamically optimize the assignment of trainable parameters. The overall architecture of PE-DyRA is shown in Figure 2. Firstly, the incremental weight matrix of LoRA layer is decomposed into SVD form, and the rank allocator performs a bidirectional adjustment on the rank size of each layer by calculating the importance score, pruning off redundant ranks and then expanding in more critical layers, and finally performing a final warmup. To prevent the adaptive capacity from being completely pruned, the minimum rank is set to 1 to maintain minimal adaptability.

### 3.1 SVD-FORM ADAPTATION

We parameterize the weight increment in the form of singular value decomposition, and represent the incremental update of the pre-trained weight matrix as

$$\mathbf{W} = \mathbf{W}^{(0)} + \Delta = \mathbf{W}^{(0)} + \mathbf{AEB} \qquad (1)$$

where $\mathbf{A} \in \mathbb{R}^{d_1 \times r}$ and $\mathbf{B} \in \mathbb{R}^{r \times d_2}$ are learnable factor matrices, $\mathbf{E} = \mathrm{diag}(e_1, \ldots, e_r)$ is a trainable diagonal matrix of singular values.

Similar to the existing work AdaLoRA (Zhang et al., 2023b), we still adopt the concept of triples. For a LoRA layer with rank $r$, each rank's corresponding triple is treated as the fundamental unit for computing importance scores and performing rank adjustments. It can be expressed as $\mathcal{G}_i = \left(\mathbf{a}_i, e_i, \mathbf{b}_i\right)$ for $i = 1, \ldots, r$, where $\mathbf{a}_i$ and $\mathbf{b}_i$ are the $i$-th column of $\mathbf{A}$ and the $i$-th row of $\mathbf{B}$, respectively.

At step $t$, the set of triples is $\mathcal{G}^{(t)} = \{G_1^{(t)}, G_2^{(t)}, \ldots, G_{r^{(t)}}^{(t)}\}$. We compute an importance score for each triple:$S_i^{(t)} = f(\mathbf{a}_i^{(t)}, e_i^{(t)}, \mathbf{b}_i^{(t)})$, where $f(\cdot)$ integrates gradient-based and input-based sensitivities. Based on $S_i^{(t)}$, we update the triple set by pruning and expansion: $\mathcal{G}^{(t+1)} = \left(\mathcal{G}^{(t)} \setminus \mathcal{G}_{\text{pruned}}^{(t)}\right) \cup \mathcal{G}_{\text{expanded}}^{(t)}$, where $\mathcal{G}_{\text{pruned}}^{(t)}$ contains the least important triples to be removed, and $\mathcal{G}_{\text{expanded}}^{(t)}$ introduces new triples initialized for critical directions.

To maintain decomposition stability, we apply spectral regularization (Zhang et al., 2023b):

$$\mathcal{R}_{\text{orth}} = \|\mathbf{A}^\top \mathbf{A} - \mathbf{I}_r\|_F^2 + \|\mathbf{B}^\top \mathbf{B} - \mathbf{I}_r\|_F^2 \tag{2}$$

### 3.2 Parameter Importance Evaluation

#### 3.2.1 Gradient-Based Parameter Importance Estimation

Inspired by AdaLoRA (Zhang et al., 2023b) and Platon (Zhang et al., 2022), we quantify parameter sensitivity using the absolute product of weights and gradients, and apply exponential moving average (EMA) smoothing across training iterations. Due to the high variability and uncertainty, the quantification of uncertainty is also performed. The final importance score is defined as the product of smoothed sensitivity and uncertainty:

$$\begin{cases} \text{Sensitivity:} & \overline{I}_{(w_{ij})}^{(t)} = \beta_1 \overline{I}_{(w_{ij})}^{(t-1)} + (1 - \beta_1) \left|w_{ij}^{(t)} \cdot \nabla_{w_{ij}} L^{(t)}\right| \\ \text{Uncertainty:} & \overline{U}_{(w_{ij})}^{(t)} = \beta_2 \overline{U}_{(w_{ij})}^{(t-1)} + (1 - \beta_2) \left|\left|w_{ij}^{(t)} \cdot \nabla_{w_{ij}} L^{(t)}\right| - \overline{I}_{(w_{ij})}^{(t)}\right| \\ \text{Importance:} & \mathcal{S}_{(w_{ij})}^{(t)} = \overline{I}_{(w_{ij})}^{(t)} \cdot \overline{U}_{(w_{ij})}^{(t)} \end{cases} \tag{3}$$

where $\beta_1, \beta_2 \in [0, 1)$ are EMA coefficients.

**Gradient-Aware Triple Importance.** During training, we observe that the gradients of the factor matrices $A$ and $B$, which correspond to the left and right singular matrices, are typically smaller by orders of magnitude compared to those of the core matrix $E$. This indicates that the update of the core matrix plays a more critical role in the optimization process (see Figure 11 in Appendix I).

Consequently, when evaluating the importance of a triple, we do not rely on a uniform linear combination of its constituent importance scores. Instead, we introduce a **gradient-aware weighting scheme**, where the contribution of each component is scaled according to the relative magnitude of its gradient. Formally, the triple-level importance score at step $t$ is defined as

$$S_{G_i}^{(t)} = \omega_E^{(t)} \cdot \mathrm{S}(E_i) + \omega_A^{(t)} \cdot \mathrm{S}(A_i) + \omega_B^{(t)} \cdot \mathrm{S}(B_i), \tag{4}$$

where the adaptive weights are computed as

$$\omega_X^{(t)} = \frac{\|\nabla_X L^{(t)}\|_2}{\|\nabla_A L^{(t)}\|_2 + \|\nabla_E L^{(t)}\|_2 + \|\nabla_B L^{(t)}\|_2}, \quad X \in \{A, E, B\}. \tag{5}$$

This formulation ensures that components with stronger optimization impact, particularly the core matrix, receive larger weights in the triple-level importance score. As a result, the aggregated importance evaluation better reflects the actual training dynamics and guides more effective rank allocation during dynamic adjustment.

For detailed empirical results validating the proposed gradient-aware weighting scheme, please refer to Appendix F.

#### 3.2.2 Input-Based Parameter Importance Estimation

Notably, on datasets with diverse input distributions, the input activations can vary from token to token. So input activations also play an equally critical role in determining the actual output of a

neuron. The contribution to the neuron output is determined jointly by magnitude of weight and the scale of the corresponding input activation.

Motivated by pruning metrics such as Wanda (Sun et al., 2024), we extend the importance estimation to the LoRA decomposition. Let $X$ denote the input activations. Given the input representation $X \in \mathbb{R}^{N \times d}$ (where $N = $ batch_size $\times$ seq_len, and $d$ is the hidden dimension), we compute the L2 norm along the batch and token dimension for each feature dimension: $\|X_j\|_2 = \frac{1}{N} \sum_{i=1}^{N} x_{ij}^2, \quad j = 1, \ldots, d$. This preserves per-feature energy, reflecting the relative contribution of each input dimension. Then we apply exponential moving average(EMA) to this. For $A \in \mathbb{R}^{d_1 \times r}$, the importance of each element is expressed as $S_{ij} = |W_{ij}| \cdot \|X_j\|_2$. The importance score for each triplet is defined as

$$S_{G_i}^{inp} = |e_i| \cdot \sum_{k=1}^{d_1} S_{ki} \cdot \|\mathbf{B}_{i:}\|_2, \quad i = 1, \ldots, r, \tag{6}$$

As shown in Algorithm 1 in Appendix I.3, the procedure of computing the overall importance score is illustrated.

**Layer-level Importance.** For a LoRA layer with $r$ ranks, corresponding to triples $\{G_1, G_2, \ldots, G_r\}$, the layer-level importance score is defined as

$$S_{\text{layer}} = \frac{1}{r} \sum_{i=1}^{r} S(G_i), \tag{7}$$

where $S(G_i)$ denotes the importance score of the $i$-th triple at the rank-level.

### 3.3 BIDIRECTIONAL RANK ADJUSTMENT STRATEGY

**Pruning Phase.** During the pruning phase, we first compute the importance score for each rank-level triple $G = (A, E, B)$. All triples are sorted according to their importance scores, and the $k$ triples with the lowest aggregated scores are selected for removal: $\mathcal{P} = \text{argmin}_{\mathcal{S}, |\mathcal{S}|=k} \sum_{G \in \mathcal{S}} S(G)$, where $\mathcal{P}$ denotes the set of pruned ranks.

**Expansion Phase.** In the expansion phase, we perform layer-level importance evaluation by aggregating rank-level scores. Based on these layer-level scores, we execute a global ranking across all layers and adopt a strict rank-conservation strategy: the $k$ rank resources removed in the pruning phase are reassigned to the top-$k$ layers with the highest $S_{\text{layer}}^{(\ell)}$ values: $\mathcal{E} = \text{argmax}_{\mathcal{S}, |\mathcal{S}|=k} \sum_{\ell \in \mathcal{S}} S_{\text{layer}}^{(\ell)}$, where $\mathcal{E}$ denotes the set of expanded layers.

Under the constraint of a fixed total rank budget, the bidirectional adjustment strategy removes less important redundant parameters and reallocates them to more critical LoRA layers, thereby improving parameter efficiency and enhancing model performance.

For detailed bidirectional rank adjustment strategy, please refer to Algorithm 2 in the appendix I.3.

### 3.4 PARAMETRIC EFFICIENCY ANALYSIS

**Theorem 3.1** (Pareto-Optimal Parameter Efficiency under Rank Allocation Constraints). *Consider $L$ LoRA layers, each assigned a rank $r_l$, under a fixed total rank budget $R_{\text{total}}$: $\sum_{l=1}^{L} r_l = R_{\text{total}}$. Let $G_l$ denote the importance score of layer $l$. The necessary condition for Pareto-optimal parameter efficiency is:*

$$r_l \propto G_l^{2/3} \tag{8}$$

*That is, layers with higher importance scores should be assigned more ranks.*

**Inference: Dynamic vs. Static Strategy** A dynamic rank adjustment strategy that updates $r_l$ in response to changes in $G_l$ during training can iteratively approach the Pareto-optimal condition equation 8. Static strategies, which fix $\{r_l\}$ at initialization, cannot adapt to evolving layer importance, and thus are generally less efficient in parameter utilization and model performance. See Appendix B for the complete derivation.

Table 2: Performance comparison of different PEFT methods on GLUE benchmark (rank $r = 8$).

| Method | SST-2 Acc. | MNLI Acc. | CoLA Mcc. | QNLI Acc. | MRPC Acc. | QQP Acc. | RTE Acc. | STS-B Corr. | All Avg. |
|--------|-----------|-----------|-----------|-----------|-----------|----------|----------|-------------|----------|
| LoRA | 95.18 | 89.74 | 69.33 | 93.90 | 89.70 | 91.99 | 86.28 | 91.66 | 88.473 |
| PiSSA | 95.53 | 90.30 | 71.41 | 94.07 | 90.20 | 91.92 | **88.09** | 91.54 | 89.133 |
| LoRA+ | 95.3 | 90.28 | 70.25 | 94.01 | 90.93 | 92.09 | 86.28 | 91.54 | 88.835 |
| AdaLoRA | 95.53 | 90.50 | 69.02 | 94.42 | 90.93 | 92.03 | 87.00 | 91.77 | 88.9 |
| DyLoRA | 95.18 | 89.51 | 69.82 | 94.29 | 89.95 | 91.97 | 85.92 | 91.74 | 88.547 |
| IncreLoRA | 95.72 | 90.62 | 70.20 | 94.36 | 90.11 | 91.91 | 86.88 | 91.38 | 88.898 |
| RandLoRA | **95.98** | 89.96 | 68.22 | 93.74 | 90.69 | 92.06 | 86.28 | 91.34 | 88.534 |
| TriAdaptLoRA | 95.68 | **90.64** | **71.6** | 94.37 | 90.77 | 92.09 | 87.84 | 91.79 | 89.348 |
| PE-DyRA | **95.98** | 90.38 | 71.43 | **94.53** | **91.18** | **92.14** | **88.09** | **91.98** | **89.464** |

## 4 EXPERIMENTS

### 4.1 MODELS AND DATASETS

**Natural Language Understanding (NLU).** We adopt **DeBERTa-v3-base** (He et al., 2021) and fine-tune it on the **GLUE benchmark** (Wang et al., 2019), using eight tasks from the benchmark.

**Mathematical Reasoning and Code Generation.** We employ **LLaMA-2-7B** (Touvron et al., 2023) and **LLaMA-3-8B** (Dubey et al., 2024) for evaluation on **mathematical reasoning**, where the models are fine-tuned on MetaMathQA (Yu et al., 2024) and assessed on GSM8K (Cobbe et al., 2021) and MATH (Hendrycks et al., 2021). For **code generation**, the models are fine-tuned on CodeFeedback (Zheng et al., 2024) and evaluated on HumanEval (Chen et al., 2021) and MBPP (Austin et al., 2021).

**Summarization.** We use **BART-large** (Lewis et al., 2019) for summarization on **XSum** (Narayan et al., 2018), which evaluate the ability to generate concise and faithful summaries.

### 4.2 BASELINES

We compare our method against a broad range of parameter-efficient fine-tuning (PEFT) approaches, including **LoRA** and its variants, as well as **dynamic rank adaptation methods**:

- **LoRA-based methods:** LoRA (Hu et al., 2022), LoRA+ (Hayou et al., 2024)
  , PiSSA (Meng et al., 2024), DoRA (Liu et al., 2024), RandLoRA (Albert et al., 2025), QLoRA (Dettmers et al., 2023), RaSA (He et al., 2025).
- **Dynamic rank methods:** AdaLoRA (Zhang et al., 2023b), IncreLoRA (Zhang et al., 2023a), DyLoRA (Valipour et al., 2023), TriAdaptLoRA (Liang et al., 2025).

### 4.3 RESULTS

**Natural Language Understanding.** Table 2 reports the performance of different PEFT methods on eight tasks from the GLUE benchmark with rank of $r = 8$. Overall, our method consistently outperforms existing baselines, achieving the highest average score (89.464%), demonstrating superior generalization across both sentence-level and sentence-pair classification tasks. For example, on the MRPC task, our method achieves 91.18% accuracy, which is 0.25% higher than the best-performing baseline (AdaLoRA, 90.93%).

**Mathematical Reasoning and Code Generation.** As shown in Table 3, our method (PE-DyRA) achieves the best performance on both LLaMA2-7B and LLaMA3-8B. On LLaMA2-7B, the overall average score of 34.69% surpasses the strongest baseline DoRA (33.99%) by nearly +0.7 points. On LLaMA3-8B, PE-DyRA also delivers the best average score (65.09%), demonstrating consistent advantages in both mathematical reasoning and code generation.

Table 3: Performance comparison of different PEFT methods on LLaMA2-7B and LLaMA3-8B.

| Model | Method | #Params(%) | GSM8K | MATH | HumanEval | MBPP | Avg |
|-------|--------|-----------|-------|------|-----------|------|-----|
| | LoRA | 0.15 | 52.90 | 7.60 | 26.0 | 34.7 | 30.3 |
| | DoRA | 0.17 | 56.09 | 9.76 | 31.7 | 38.4 | 33.99 |
| LLaMA2-7B | QLoRA | 0.28 | 50.32 | 6.12 | 24.8 | 32.8 | 28.51 |
| | RaSA | 0.15 | **56.41** | 9.72 | 28.0 | 36.2 | 32.58 |
| | AdaLoRA | 0.22 | 52.01 | 8.26 | 26.2 | 35.2 | 30.42 |
| | PE-DyRA | 0.15 | 56.25 | **10.12** | **33.5** | **38.9** | **34.69** |
| | LoRA | 0.13 | 81.27 | 39.04 | 64.0 | 69.0 | 63.33 |
| | DoRA | 0.15 | **81.42** | 37.22 | 65.2 | 72.0 | 63.96 |
| LLaMA3-8B | QLoRA | 0.23 | 81.12 | 39.58 | 67.1 | 70.6 | 64.6 |
| | RaSA | 0.13 | 80.97 | 36.18 | 67.1 | 69.6 | 63.46 |
| | AdaLoRA | 0.20 | 81.04 | 39.62 | 65.9 | 72.0 | 64.64 |
| | PE-DyRA | 0.13 | 80.82 | **40.24** | **67.1** | **72.2** | **65.09** |

**Summarization.** Table 4 shows the results on summarization task. Compared with LoRA and AdaLoRA, our method (PE-DyRA) achieves the best performance across all Rouge metrics while using the same parameter budget as LoRA (2.06M).

Table 4: Performance comparison of different PEFT methods on XSum.

| Method | #Params | Rouge-1 | Rouge-2 | Rouge-L | Rouge-Lsum |
|--------|---------|---------|---------|---------|------------|
| LoRA | 2.06M | 43.6283 | 20.4566 | 35.6239 | 35.6194 |
| AdaLoRA | 3.09M | 43.9557 | 20.5627 | 35.6264 | 35.6129 |
| PE-DyRA | 2.06M | **44.0444** | **20.8523** | **35.9616** | **35.9602** |

## 4.4 PE-DyRA as a General Paradigm

To further validate the generality of our proposed method, we integrate PE-DyRA into several representative PEFT approaches, including PiSSA, DoRA, and QLoRA, across both natural language understanding (NLU) and code generation tasks. The results are reported in Table 11 and Table 12 in Appendix I.2.

On the DeBERTa-v3-base NLU benchmark, PE-DyRA consistently improves PiSSA, achieving an average accuracy of 89.29%. For LLaMA models on code generation, PE-DyRA also provides significant gains. On LLaMA2-7B, PE-DyRA+DoRA improves MBPP by 1.3% over DoRA. On LLaMA3-8B, PE-DyRA+QLoRA achieves 74.3% on MBPP, outperforming QLoRA by 3.7%.

These results suggest that PE-DyRA can be applied as a paradigm to existing PEFT methods, improving their performance across different models and tasks. This highlights its wide applicability and generalization ability.

## 4.5 Analysis

### 4.5.1 Ablation Study on Bidirectional Strategies

We propose a bidirectional rank adjustment strategy and validate it via an ablation study. From Table 5, compared with Prune-only or Expand-only variants, our method (PE-DyRA) balances parameter allocation and model capacity, achieving superior performance without increasing the total number of training parameters.

Table 5: Performance comparison with different strategies.

| Method | MRPC | STS-B | RTE | CoLA | SST-2 | QNLI | QQP | MNLI | Avg |
|--------|------|-------|-----|------|-------|------|-----|------|-----|
| LoRA | 89.70 | 91.66 | 86.28 | 69.33 | 95.18 | 93.90 | 91.99 | 89.74 | 88.473 |
| +Prune | 90.20 | 91.80 | 87.36 | 70.19 | 95.53 | 94.34 | 92.13 | 90.37 | 88.99 |
| +Expand | 90.69 | 91.72 | **88.09** | 70.21 | 95.30 | 94.51 | **92.46** | 90.30 | 89.16 |
| PE-DyRA | **91.18** | **91.98** | **88.09** | **71.43** | **95.98** | **94.53** | 92.14 | **90.38** | **89.464** |

### 4.5.2 ABLATION OF INPUT-BASED IMPORTANCE

To investigate the effect of input-based importance calculation in PE-DyRA, we conduct an ablation study on LLaMA2-7B across code generation benchmarks. We conducted experiments on two ways, without input-based importance and with input-based importance, and analyzed the results.

Table 6: Ablation study on input-based importance calculation in PE-DyRA.

| Model | Variant | Humaneval | Humaneval+ | MBPP | MBPP+ |
|-------|---------|-----------|------------|------|-------|
| LLaMA2-7B | w/o input-based | 32.3 | 29.3 | 38.4 | 29.9 |
| | with input-based | **33.5** | **30.5** | **38.9** | **32.5** |

As shown in Table 6, incorporating input-based importance calculation improves performance on benchmarks, highlighting its effectiveness in PE-DyRA.

### 4.5.3 LAYER-WISE RANK DYNAMICS

To analyze the dynamic rank adjustment mechanism, we track the evolution of allocated ranks and corresponding importance rankings for representative layers during training (Figure 3, (a) rank dynamics; (b) importance ranking, where smaller values denote higher importance).

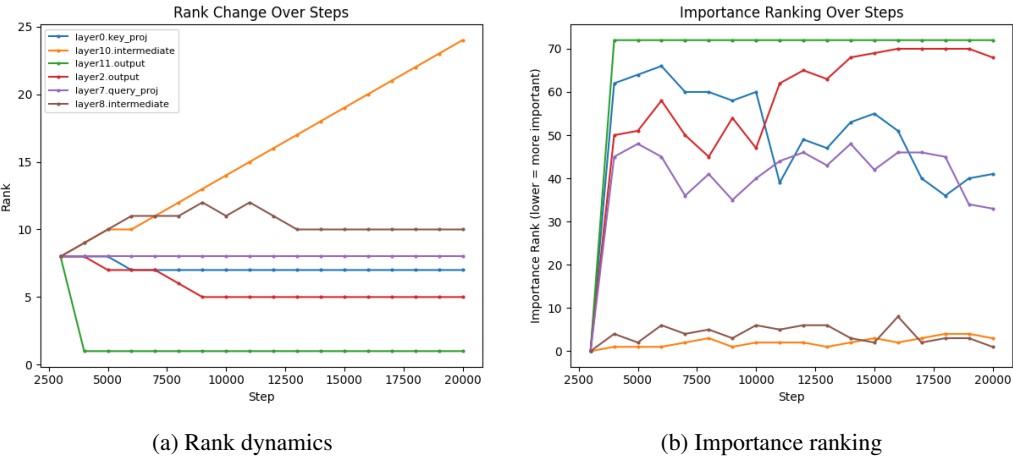

(a) Rank dynamics        (b) Importance ranking

Figure 3: Evolution of dynamic rank allocation and layer importance during training.

We observe clear differences across layers. `layer10.intermediate` consistently maintains high importance throughout training, leading to an increase in its allocated rank. In contrast, `layer11.output` exhibits low importance and rapidly reduces its rank at the beginning of training. Other layers show fluctuating importance rankings, resulting in relatively stable rank changes. Notably, `layer8.intermediate` remains among the top in importance ranking, causing its rank to increase at each update; however, the number of its ranks fluctuates up and down over time. This suggests that certain triples at the rank level have low importance and are thus pruned.

### 4.5.4 PERFORMANCE UNDER DIFFERENT RANK BUDGETS

As shown in Table 13 in Appendix I.2, we also test the fine-tuning performance of the proposed method on some datasets with different rank budgets. It can be observed that the proposed method achieves performance improvement under different budgets.

### 4.5.5 ABLATION STUDY ON RANK ADJUSTMENT SIZE

Our method performs rank updates every $T$ steps, where the adjustment size (number of ranks pruned/expanded) critically affects performance. We evaluate adjustment sizes $\{4, 8, 12, 24\}$ on DeBERTaV3-base with initial rank $r = 8$ to analyze this effect.

As shown in Table 14 in Appendix I.2, when the dynamic rank adjustment is small, the model's ability to improve is limited; when it is large, pruning trained ranks and introducing new ones can destabilize training. The optimal adjustment value depends on both the model architecture and the initial rank size.

### 4.5.6 IMPORTANCE-DRIVEN CAUSAL TESTING AND SPEARMAN CORRELATION

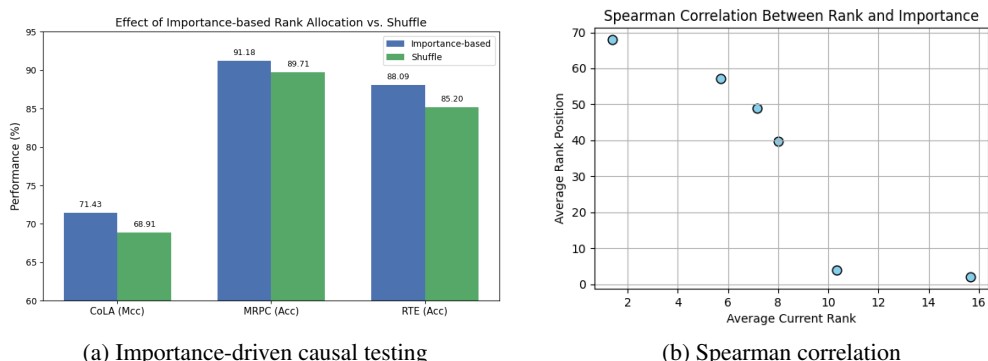

(a) Importance-driven causal testing     (b) Spearman correlation

Figure 4: Effectiveness of importance-guided dynamic rank allocation. (a) Causal test comparing importance-based allocation with random shuffle. (b) Spearman correlation between mean allocated rank and mean importance ranking.

In Figure 4a, we test causality by randomizing the importance order during training, making rank adjustment random. The resulting performance drop across datasets confirms that importance-to-rank assignment is indeed effective. In Figure 4b, we analyze the correlation between layer importance and assigned rank. The strong Spearman correlation confirms that dynamic rank assignment aligns well with learned importance.

## 5 CONCLUSION AND FUTURE WORK

This study proposes PE-DyRA, an efficient dynamic rank adjustment method that improves parameter utilization. Experiments demonstrate that PE-DyRA outperforms existing fine-tuning approaches across diverse tasks, validating its effectiveness for large-scale models under limited resources.

However, there is still much future work to be done in this research. The size of the adjusted rank in the update is currently fixed as a parameter, and it can be extended to an adaptive method to determine the size of the adjusted rank in the update independently. In the future, we will explore more appropriate measures of importance, apply our method to larger models, and extend it to various tasks such as federation, multi-task, and domain adaptation. These promising challenges remain to be explored in future research efforts.

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

# A    THE USE OF LARGE LANGUAGE MODELS

A large language model (LLM) was used to assist in refining the writing style and polishing the language of this paper. We gratefully acknowledge its contribution in improving the readability and clarity of the manuscript. All LLM-generated content was reviewed and corrected by the authors to maintain accuracy and preserve the original meaning.

# B    PROOF OF PARETO-OPTIMAL PARAMETER EFFICIENCY

We formalize the rank allocation problem under a fixed total rank budget $R_{\text{total}}$ using a Lagrangian framework. Consider $L$ LoRA layers, each assigned a rank $r_l > 0$, with layer importance scores $G_l > 0$. We assume that the layer-wise contribution to the overall loss can be approximated as

$$L(\{r_l\}) = \sum_{l=1}^{L} \frac{G_l}{\sqrt{r_l}}, \tag{9}$$

subject to the total rank budget constraint

$$\sum_{l=1}^{L} r_l = R_{\text{total}}. \tag{10}$$

Our goal is to minimize equation 9 subject to equation 10, yielding the most parameter-efficient rank allocation.

**Lagrangian formulation.**    We construct the Lagrangian

$$\mathcal{J}(\{r_l\}, \lambda) = \sum_{l=1}^{L} G_l r_l^{-1/2} + \lambda \left( \sum_{l=1}^{L} r_l - R_{\text{total}} \right), \tag{11}$$

where $\lambda$ is the Lagrange multiplier for the total rank constraint.

**Optimality condition.**    Taking the derivative of equation 11 with respect to $r_l$ and setting it to zero for optimality, we obtain

$$\frac{\partial \mathcal{J}}{\partial r_l} = -\frac{1}{2} G_l r_l^{-3/2} + \lambda = 0. \tag{12}$$

Equation equation 12 implies that

$$G_l r_l^{-3/2} = 2\lambda, \quad \forall l. \tag{13}$$

Since the right-hand side is independent of $l$, we have

$$r_l^{3/2} \propto G_l \implies r_l \propto G_l^{2/3}. \tag{14}$$

**Global optimality.**    Each term $G_l r_l^{-1/2}$ is strictly convex in $r_l > 0$, so the total objective equation 9 is strictly convex, and the constraint equation 10 is linear. Therefore, any stationary point satisfying equation 13 is the unique global minimizer. Thus, this is the globally Pareto-optimal rank allocation.

**Implications.**    Compared to any static allocation (uniform $r_l = R_{\text{total}}/L$), the dynamic allocation achieves a strictly lower loss in equation 9 whenever the importance scores $G_l$ are not all equal. This formally establishes that allocating ranks proportionally to $G_l^{2/3}$ is Pareto-optimal under the given model.

For the SST-2 dataset, both LoRA(using SVD triples) and PE-DyRA methods are used to verify the above results, and the following graphs are plotted: The horizontal axis is the step during the training process; The vertical axis is the log variance of $G_l^{2/3}/r_l$ across layers (which indicates how much the value deviates from the constant across layers).

As shown in Figure 5, under the dynamic strategy, the cross-layer variance of $G_l^{2/3}/r_l$ gradually decreases during training, indicating that the model progressively approaches the Pareto optimal condition. Moreover, the variance under the dynamic strategy is consistently lower than that of the static LoRA strategy, suggesting that the dynamic rank adjustment achieves a more Pareto-efficient parameter allocation.

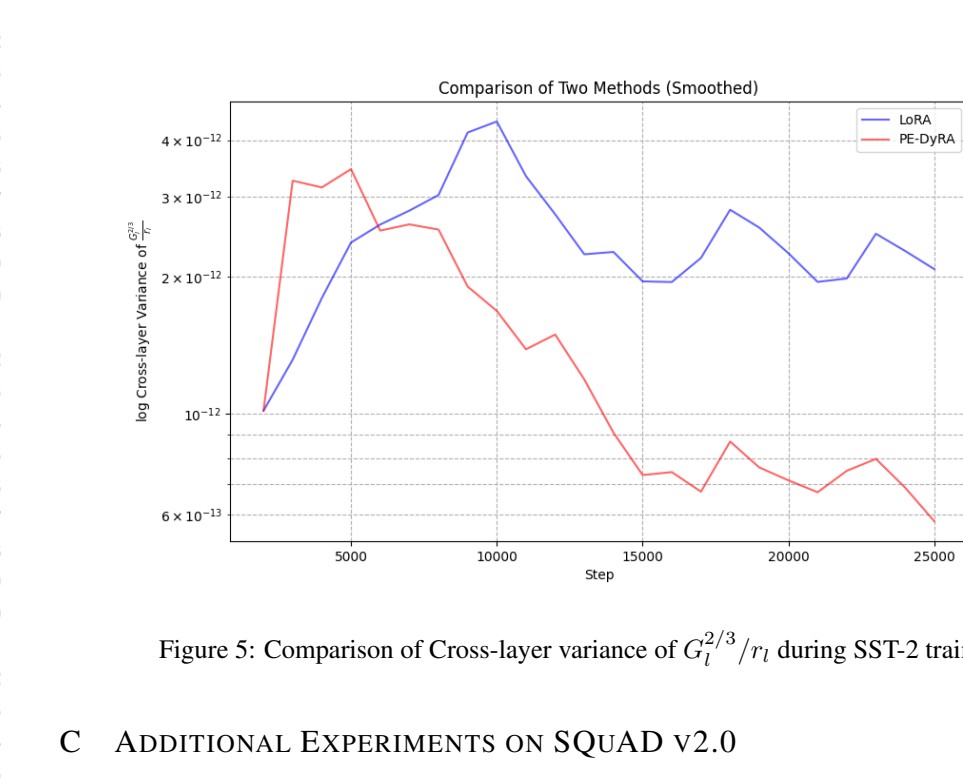

Figure 5: Comparison of Cross-layer variance of $G_l^{2/3}/r_l$ during SST-2 training.

## C  ADDITIONAL EXPERIMENTS ON SQUAD V2.0

To further illustrate the performance difference between LoRA and PE-DyRA, we provide a bar chart comparison of representative evaluation metrics, including HasAns F1, NoAns F1, Exact Match, and Overall F1 on SQuAD v2.0.

As shown in Figure 6, PE-DyRA matches LoRA on HasAns F1 while substantially improving NoAns F1, leading to higher Exact Match and Overall F1. This demonstrates that PE-DyRA maintains strong performance on answerable questions and enhances robustness on unanswerable ones.

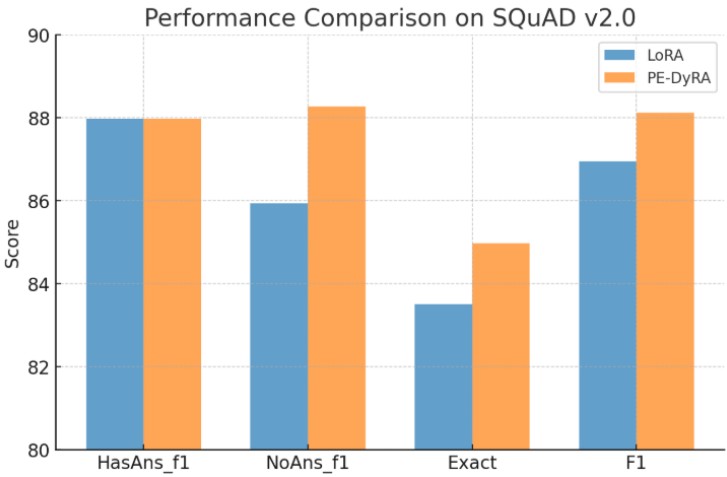

Figure 6: Comparison of evaluation metrics between LoRA and PE-DyRA on SQuAD v2.0.

## D  THE RESULTING RANK DISTRIBUTION

Methods were applied to SST-2 using the DeBERTaV3-base model and the respective final rank distributions were saved.

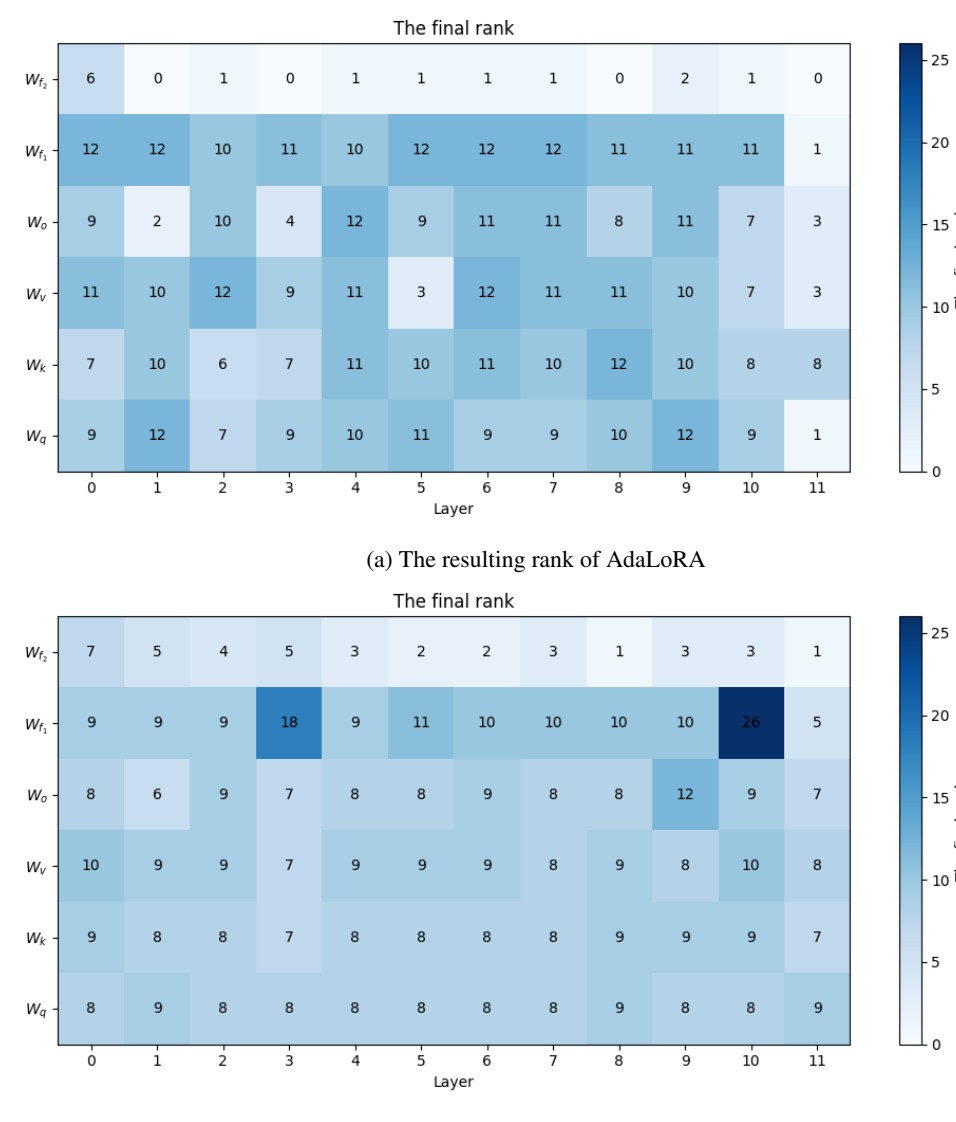

(a) The resulting rank of AdaLoRA

(b) The resulting rank of PE-DyRA

Figure 7: Comparative rank allocation patterns across model layers.

As shown in Figure 7, in the case of limited resources, the proposed method tends to produce relatively concentrated rank distributions. This may be one of the reasons why it is more effective in low-resource settings: by prioritizing assigning higher ranks to critical modules, methods are able to maintain adequate representation of important modules even with a limited parameter budget, thus achieving superior overall performance.

# E EXPERIMENTAL SETTINGS

## E.1 TRAINING DETAILS ON GLUE BENCHMARK

In the GLUE Benchmark, the model we used was DebertaV3-Base, with a rank size of 8. The specific details of the experimental hyperparameters are shown in the table 7.

Table 7: Hyper-parameter setup of PE-DyRA for GLUE benchmark.

| Dataset | learning rate | batch size | # epochs | $\gamma$ | $t_i$ | $\Delta_T$ | $t_f$ | $k$ |
|---------|---------------|------------|----------|----------|-------|------------|-------|-----|
| MNLI | $5 \times 10^{-4}$ | 32 | 7 | 0.1 | 3000 | 1000 | 65000 | 12 |
| RTE | $1.2 \times 10^{-3}$ | 32 | 50 | 0.3 | 300 | 100 | 2600 | 12 |
| QNLI | $9 \times 10^{-4}$ | 32 | 5 | 0.1 | 1000 | 500 | 10000 | 12 |
| MRPC | $1 \times 10^{-3}$ | 32 | 30 | 0.1 | 600 | 150 | 1100 | 12 |
| QQP | $6 \times 10^{-4}$ | 32 | 9 | 0.1 | 5000 | 1000 | 80000 | 12 |
| SST-2 | $8 \times 10^{-4}$ | 32 | 24 | 0.1 | 1000 | 1000 | 25000 | 12 |
| CoLA | $1 \times 10^{-3}$ | 32 | 35 | 0.1 | 700 | 100 | 7000 | 12 |
| STS-B | $2.2 \times 10^{-3}$ | 32 | 25 | 0.3 | 800 | 200 | 1500 | 12 |

Table 8: Hyper-parameter setup of PE-DyRA for mathematical reasoning and code generation.

| Model | Dataset | learning rate | batchsize | # epochs | $t_i$ | $\Delta_T$ | $t_f$ | $k$ |
|-------|---------|---------------|-----------|----------|-------|------------|-------|-----|
| LLaMA2-7B | MetaMath | $2 \times 10^{-4}$ | 16 | 5 | 2000 | 1000 | 11250 | 12 |
| | Python | $2 \times 10^{-4}$ | 16 | 5 | 2000 | 1000 | 12765 | 12 |
| LLaMA3-8B | MetaMath | $5 \times 10^{-5}$ | 16 | 5 | 2000 | 1000 | 11250 | 12 |
| | Python | $1 \times 10^{-4}$ | 16 | 5 | 2000 | 1000 | 12765 | 12 |

### E.2 TRAINING DETAILS ON MATHEMATICAL REASONING AND CODE GENERATION

In the Mathematical Reasoning and Code Generation task, the LLaMA2-7B and LLaMA3-8b models were used, with an initial rank size of 4. The specific details of the experimental hyperparameters are shown in the table 8.

### E.3 TRAINING DETAILS ON SUMMARIZATION AND QA

Table 9: Hyper-parameter setup of PE-DyRA for summarization and QA.

| Dataset | learning rate | batch size | # epochs | $\gamma$ | $t_i$ | $\Delta_T$ | $t_f$ | $k$ |
|---------|---------------|------------|----------|----------|-------|------------|-------|-----|
| XSum | $2 \times 10^{-4}$ | 24 | 25 | 0.1 | 6000 | 1500 | 180000 | 12 |
| SQuAD v2.0 | $1.2 \times 10^{-3}$ | 16 | 25 | 0.1 | 5000 | 1000 | 190000 | 12 |

For the summary and question-answering tasks, the XSum dataset uses the BART-large model, while the SQuAD v2.0 (Rajpurkar et al., 2018) uses the DebertaV3-Base model. The initial rank size used is 4. The specific details of the experimental hyperparameters are shown in the table 9.

## F EMPIRICAL VALIDATION OF GRADIENT-AWARE WEIGHTING

To validate the proposed gradient-aware weighting scheme for triple importance scores, we track rank evolution during training using a temporary zero-masking strategy with initialized. As shown in Figure 8, with a simple linear combination of importance scores(Figure 8a), pruned ranks often reactivate, indicating unstable importance evaluation. In contrast, the gradient-aware scheme(Figure 8b) keeps pruned ranks consistently suppressed, better reflecting optimization dynamics and this is consistent with the effect we want.

As shown in Figure 9, we monitor the change in loss for both cases with and without using the gradient-aware weighting scheme. Through the loss change during training, it can be seen that using gradient information weighting can accelerate the optimization and make the optimization process more stable.

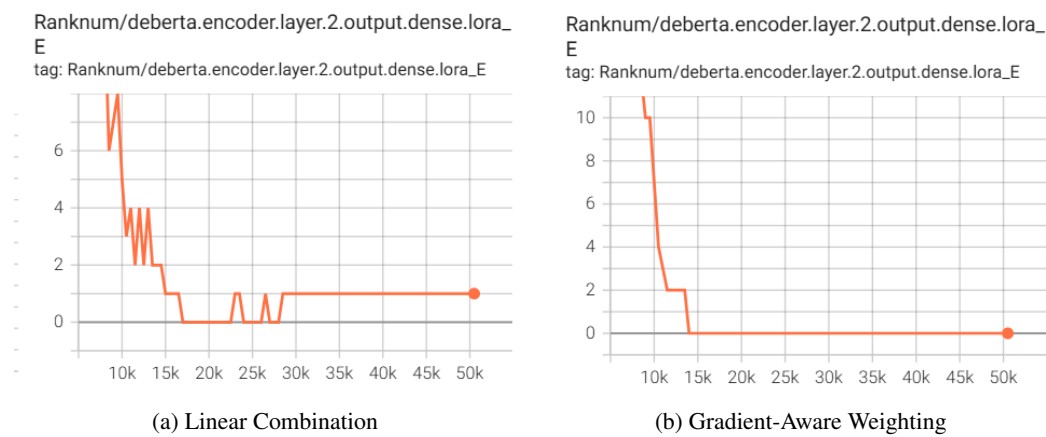

(a) Linear Combination                    (b) Gradient-Aware Weighting

Figure 8: Comparison of rank evolution under two importance aggregation schemes: (a) linear combination leads to unstable pruning with ranks repeatedly reappearing, while (b) gradient-aware weighting yields stable pruning dynamics.

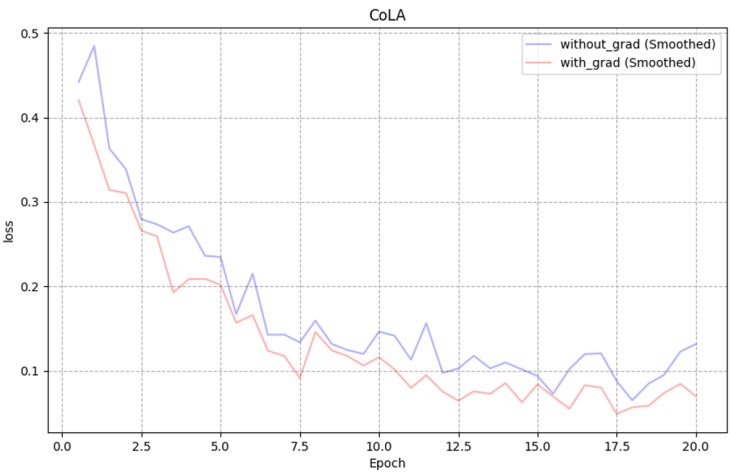

Figure 9: Training loss comparison with/without gradient weighting

## G  TIME–MEMORY–ACCURACY COMPARISON

Table 10: Comparison of training time, memory usage, and accuracy among different methods.

| Dataset | Method | #Params | Runtime/epoch(s) | Peak Memory $\Delta$ | Acc (%) |
|---------|--------|---------|------------------|----------------------|---------|
|         | LoRA    | 1.33M | 220.64 | 4306MB  | 95.18 |
| SST-2   | AdaLoRA | 1.99M | 403.09 | 4321MB  | 95.53 |
|         | PE-DyRA | 1.33M | 341.22 | 4315MB  | 95.98 |
|         | LoRA    | 1.33M | 31.40  | 11439MB | 89.70 |
| MRPC    | AdaLoRA | 1.99M | 38.99  | 11461MB | 90.93 |
|         | PE-DyRA | 1.33M | 37.23  | 11446MB | 91.18 |
|         | LoRA    | 1.33M | 21.65  | 11439MB | 86.28 |
| RTE     | AdaLoRA | 1.99M | 26.65  | 11461MB | 87.00 |
|         | PE-DyRA | 1.33M | 26.79  | 11446MB | 88.09 |

As shown in Table 10, our proposed PE-DyRA achieves consistently higher accuracy than LoRA and AdaLoRA while maintaining comparable parameter scale and memory usage. Notably, PE-DyRA

substantially reduces runtime (e.g., 220.64s vs. 341.22s on SST-2), demonstrating the effectiveness of dynamic rank adjustment.

## H ORTHOGONALITY REGULARIZATION LOSS

In the main text, we introduced the orthogonality regularization term, as defined in equation 2, which encourages the low-rank factors $\mathbf{A}$ and $\mathbf{B}$ to remain close to an orthogonal basis, thereby stabilizing training. The overall training objective can then be written as: $\mathcal{L}_{\text{total}} = \mathcal{L}_{\text{train}} + \lambda \mathcal{R}_{\text{orth}}$, where $\mathcal{L}_{\text{train}}$ denotes the standard training loss, and $\lambda$ is the regularization coefficient.

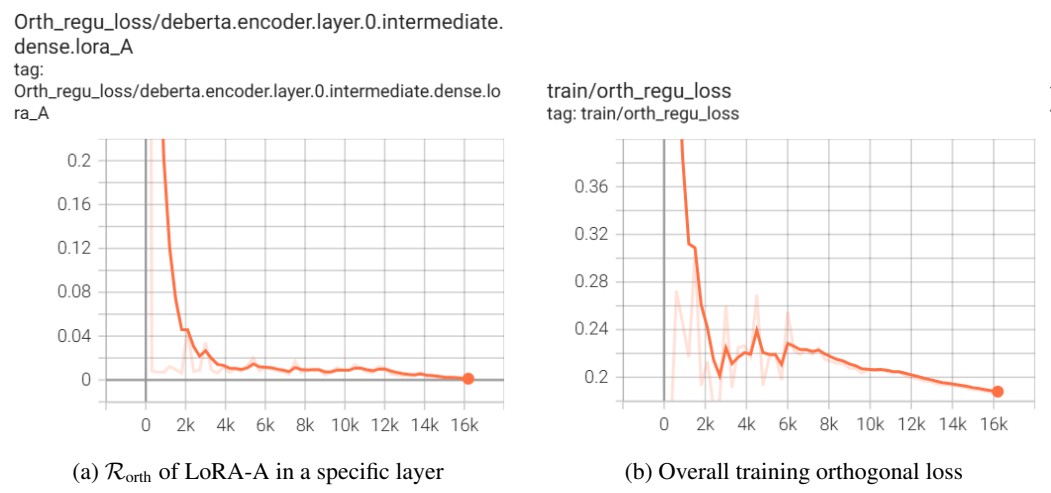

(a) $\mathcal{R}_{\text{orth}}$ of LoRA-A in a specific layer      (b) Overall training orthogonal loss

Figure 10: Dynamics of orthogonal regularization during training.

Figure 10 shows that the orthogonal loss decreases rapidly within each layer early in training and then stabilizes, indicating effective local enforcement of orthogonality. Globally, despite layer-wise fluctuations, the model consistently maintains orthogonality throughout training, demonstrating that the regularization stabilizes both local representations and overall low-rank adaptation.

## I ADDITIONAL FIGURES ALGORITHMS AND TABLES

### I.1 FIGURES

Figure 11 is mentioned in Section 3.2.1.

### I.2 TABLES

Table 11: DeBERTa-v3-base NLU benchmark results.

| Method | QNLI | MRPC | QQP | STS-B | MNLI | SST-2 | CoLA | RTE | Avg |
|---|---|---|---|---|---|---|---|---|---|
| PiSSA | 94.07 | 90.20 | 91.92 | 91.54 | **90.30** | 95.53 | 71.41 | **88.09** | 89.13 |
| PE-DyRA+PiSSA | **94.36** | **90.44** | **92.25** | **91.81** | 90.22 | **95.98** | **72.28** | 87.00 | **89.29** |

Table 11 and Table 12 reports the detailed results corresponding to Section 4.4.

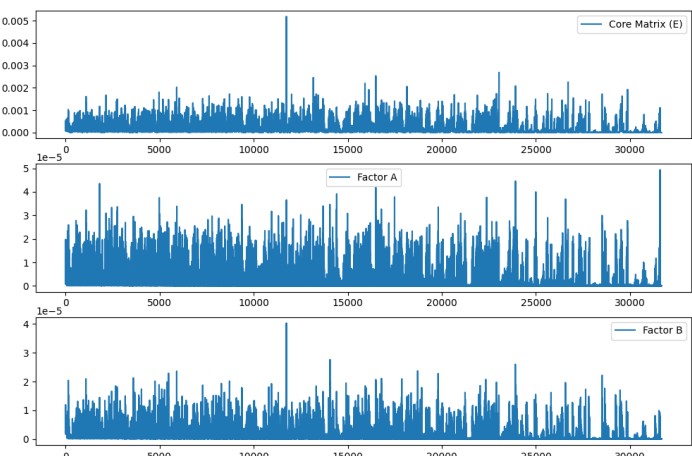

Figure 11: The gradient changes of the core matrix and factor matrix during the training process.

Table 12: LLaMA models code generation results.

| Model | Method | Performance | |
|---|---|---|---|
| | | **MBPP** | **MBPP+** |
| LLaMA2-7B | DoRA | 38.4 | 28.3 |
| | PE-DyRA+DoRA | **39.7** | **31.2** |
| | QLoRA | 32.8 | 27.0 |
| | PE-DyRA+QLoRA | **34.7** | 27.0 |
| LLaMA3-8B | DoRA | 72.0 | 61.4 |
| | PE-DyRA+DoRA | **73.0** | **63.0** |
| | QLoRA | 70.6 | 60.6 |
| | PE-DyRA+QLoRA | **74.3** | **63.8** |

Table 13: Performance under different rank budgets.

| Dataset | Method | rank=4 | rank=8 | rank=16 | rank=32 |
|---|---|---|---|---|---|
| CoLA | LoRA | 68.57 | 69.33 | 71.03 | 69.92 |
| | PE-DyRA | **71.43** | **71.43** | **71.80** | **70.85** |
| STS-B | LoRA | 91.5 | 91.66 | 91.62 | 91.52 |
| | PE-DyRA | **91.83** | **91.98** | **91.81** | **91.98** |
| MRPC | LoRA | 90.44 | 89.70 | 89.22 | 90.2 |
| | PE-DyRA | **91.18** | **91.18** | **90.2** | **90.44** |

Table 13 reports the detailed results corresponding to Section 4.5.4.

Table 14 reports the detailed results corresponding to Section 4.5.5.

## I.3 THE ALGORITHM FOR COMPUTING IMPORTANCE SCORE AND BIDIRECTIONAL RANK ADJUSTMENT STRATEGY

Table 14: Performance comparison with different rank adjustment sizes.

| Dataset | k=4 | k=8 | k=12 | k=24 |
|---|---|---|---|---|
| SST-2 (Acc.) | 95.18 | 95.52 | **95.98** | 95.18 |
| RTE (Acc.) | 85.92 | 87.36 | **88.09** | 87.84 |
| STS-B (Corr.) | 91.39 | 91.50 | **91.98** | 91.22 |

---

**Algorithm 1** Computation of Overall Triple Importance Score

---

**Require:** Parameters $\Theta = \{A, E, B\}$, input activations $X$, loss $L$
**Ensure:** Overall importance score $S$
1: **Step 1: Gradient-based triple importance**
2: Compute component-level scores $S_A^{\text{grad}}, S_E^{\text{grad}}, S_B^{\text{grad}}$ using $\nabla L$
3: **for all** $K \in \{A, E, B\}$ **do**
4:     Compute sensitivity score: $S_K^{\text{grad}} \leftarrow f_{\text{grad}}(K, \nabla_K L)$
5: **end for**
6: Fuse scores with gradient-aware weights:
$$S_{G_i}^{\text{grad}} \leftarrow \omega_A S_{A_i}^{\text{grad}} + \omega_E S_{E_i}^{\text{grad}} + \omega_B S_{B_i}^{\text{grad}}, \text{ where } \omega_K = \frac{\|\nabla_K L\|}{\sum_{H \in \{A,E,B\}} \|\nabla_H L\|}$$
7: **Step 2: Input-based triple importance**
8: Compute input-based scores $S^{\text{inp}}$ from $X$
$$S_{G_i}^{inp} = |e_i| \cdot \sum_{k=1}^{d_1} S_{ki} \cdot \|\mathbf{B}_{i:}\|_2, \quad i = 1, \ldots, r,$$
9: **Step 3: Final aggregation**
10: Compute overall score:
$$S_{G_i} = \alpha \cdot S_{G_i}^{\text{grad}} + (1 - \alpha) \cdot S_{G_i}^{\text{inp}}, \text{ where } \alpha \text{ is appropriately chosen within } [0.0, 1.0].$$
11: **return** $S$

---

**Algorithm 2** Bidirectional Rank Adjustment Strategy

---

1: **Input:** LoRA layers with rank-level triples $\{G_i = (A_i, E_i, B_i)\}$, total rank $r$, pruning size $k$
2: **Output:** Updated low-rank matrix $\Delta W_{\text{adjusted}}$
3:
4: **Step 1: Compute importance scores**
5:     Compute rank-level importance scores $S(G_i)$ for all triples $G_i$
6:     (computation procedure detailed in Algorithm 1).
7:
8: **Step 2: Pruning Phase**
9:     Select the $k$ triples with the lowest importance scores:
10:     $\mathcal{P} = \text{argmin}_{\mathcal{S}, |\mathcal{S}|=k} \sum_{G \in \mathcal{S}} S(G)$
11:     Retain the remaining triples:
12:     $\Delta W_{\text{pruned}} = \sum_{i \in \mathcal{K}} r_i, \quad \mathcal{K} = \{i \mid i \notin \mathcal{P}\}$
13:
14: **Step 3: Expansion Phase**
15:     For each layer $\ell$, compute layer-level importance score:
16:     $S_{\text{layer}}^{(\ell)} = \frac{1}{r_\ell} \sum_{i=1}^{r_\ell} S(G_i)$
17:     Select top-$k$ layers for expansion:
18:     $\mathcal{E} = \text{argmax}_{\mathcal{S}, |\mathcal{S}|=k} \sum_{\ell \in \mathcal{S}} S_{\text{layer}}^{(\ell)}$
19:     Expand ranks on selected layers:
20:     $\Delta W_{\text{adjusted}} = \Delta W_{\text{pruned}} + \sum_{j \in \mathcal{E}} r_j^{\text{new}}$
21:
22: **return** $\Delta W_{\text{adjusted}}$

---

