# OpenReview forum: "PE-DyRA: Dynamic Rank Adaptation for Parameter-Efficient  Fine-Tuning via Importance-Aware Pruning and Expansion"
_ICLR.cc/2026/Conference — ICLR 2026 Conference Withdrawn Submission_

### Official Review · Reviewer_joZZ · 2025-10-18

**Soundness:** 2
**Presentation:** 2
**Contribution:** 2
**Rating:** 4
**Confidence:** 4

**Summary:**

The method is a variant of LoRA that parameterize the adapter as AEB (instead of AB) in the form of SVD (similar to AdaLoRA) and then the ranks of the layers are reduced or increased based on gradients, weights, and input activations.

**Strengths:**

* Interesting and logical method
* Good results on Llama 2,3

**Weaknesses:**

* Method is much slower in comparison to LoRA (Table 10)
* Table 2: Did you copy the baseline results or run them yourselves? If the latter, what hyperparams have you used? What seeds? How many runs?
* Table 2: You wrote that AdaLoRA got 88.90 but in their paper they say 89.31. It’s a big difference.
* Table 2: There is no stdev. I think the bare minimum is stdev at least for your method, for consecutive seeds: 1,2,3. We need to see the seed sensitivity, as it can be something around half a point. This should be validated using the supplied code.
* Table 2: In table 3, with Llama2, the method was compared against DoRa, and it
was the strongest competitor. For some reason, this competitor is omitted from this table.
This raises the suspicion that it was omitted here since it outperformed the proposed method?
* Table 3: There is no information in the paper about the hyperparams of the baseline methods, and how they were selected. Hyper params are very important for the performance and the comparison should be fair. How can the reader know you invested the same energy/methodology in finding optimal hypers for baseline methods like you did for your method?
* Table 3: Seed sensitivity or stdev is missing. We know from the original LoRA paper that seeds can have a significant effect (sometimes even 0.7), so it is also important on Llama, at least for your method.
* Theorem, equation (9): How did you get to this assumption? Why is it logical? This assumption directly affects goal of the theorem.
* Table 10: Per dataset, all methods run the same number of epochs? if not, total training time should be specified for fair comparison.
* Line 918: It says: “substantially reduces runtime (e.g., 220.64s vs. 341.22s on SST-2)”, but Table 10 shows exactly the opposite numbers.

**Questions:**

* abstract, line 24, space is missing after 3)
* Line 46, space is missing before AdaLoRA
* Table 2: Better to keep the same number of decimal places for readability (see 88.9 and 71.6).
* Table 2, I suggest to put the model name (Deberta) in the table title, so that tables will be as self explanatory as possible and to avoid confusion. Table 3 mentioned the model type, but table 2 does not. Additionally, when you refer to the table, you should say something about the hyperparameters of the baseline models, as they are very important.
* Introduction, line 53, “often neglecting input activations": there are methods, like PRiLoRA that takes into account weights and activations like Wanda.
* Table 3: keep the same number of decimal places in all places, for readability.
* Table 5: Here you mix two decimal places with three decimal places.
* Line 809: You show the hyperparams of your method, but say something about the hyperparams of the other methods for completeness.

---

### Official Review · Reviewer_KJpm · 2025-10-28

**Soundness:** 2
**Presentation:** 1
**Contribution:** 2
**Rating:** 2
**Confidence:** 4

**Summary:**

This paper proposes PE-DyRA, a dynamic rank adjustment method for LoRA-based parameter-efficient fine-tuning. The approach introduces a rank allocator (core matrix) component and integrates gradient- and input-based importance estimation to guide bidirectional rank pruning and expansion.

**Strengths:**

- The idea of integrating input activations and gradient information into the importance estimation is conceptually interesting and may contribute to more stable rank evaluation.
- The paper provides experiments across multiple benchmarks and some ablation study, suggesting PE-DyRA outperforms existing baselines.

**Weaknesses:**

- The writing and technical presentation require major revision.

  - Several symbols are inconsistent, ambiguous, or reused, making the paper hard to follow.
  - Frequent inconsistent capitalization, and irregular precision across tables.

  - The two importance metrics share the same notation S(⋅) without clear distinction or explanation. Moreover, the aggregation step described in Algorithm 1 does not appear in the main text.
  - Key parameters such as $t_i,ΔT,t_f$ are never explained.

  - Some expressions are not sufficiently academic
    - e.g., “batch size × seq len” in Line 220; “, j = 1,…, d” in Line 222.

- Reproducibility concerns: Many hyperparameters are not specified or ablated. The paper lacks error bars and variance statistics, preventing proper assessment of robustness.
- Reported gains over existing methods (e.g., AdaLoRA, DoRA) are minor, while the proposed method introduces additional computation.

**Questions:**

- Could the authors show the distribution of the adaptive weights in Eq. 5? Line 195 states that “the core matrix plays a more critical role”; does this imply that the other two components receive negligible weights? Are there performance comparisons in Appendix F?
- How are the hyperparameters α, β, λ chosen? The paper provides no sensitivity analysis. What does “appropriately chosen” mean in practice?
- Could the assumption in Eq. 9 be further justified or empirically supported?
- How to compute the effective rank in this paper? This detail is not explained.
- Lines 223–224 vaguely describe “per-feature energy” and the EMA process. Please specify these computations clearly.
- In LLaMA-2/3, feed-forward and attention matrices differ in dimension. Thus, pruning or expansion could change the parameter count even if rank stays constant. Are the reported table values consistent with this difference?

---

### Official Review · Reviewer_Xg83 · 2025-10-30

**Soundness:** 3
**Presentation:** 3
**Contribution:** 2
**Rating:** 4
**Confidence:** 3

**Summary:**

This paper introduces PE-DyRA, a novel framework for the parameter-efficient fine-tuning (PEFT) of large language models. The authors identify two key limitations in existing methods like LoRA:1)  The inefficiency of a fixed-rank allocation and 2)  The one-directional nature (either pruning or expansion) of current dynamic methods.

To address this, PE-DyRA proposes two main contributions:

### a) A bidirectional rank adjustment mechanism
At scheduled intervals, the framework prunes the least important ranks (based on an importance score) and simultaneously reallocates (expands) that same rank budget to the most important layers. This maintains a constant total parameter budget while dynamically redistributing resources.

### b) A composite importance score
To guide the adjustment, the paper proposes a new importance metric that combines two sources of information:

1.  A **gradient-based** score, which (like AdaLoRA) uses the product of smoothed sensitivity and uncertainty.
2.  A novel **input-based** score, which incorporates the magnitude of input activations (inspired by the Wanda pruning metric).

The method utilizes the same SVD-based parameterization ($\Delta=AEB$) as AdaLoRA to enable rank-level adjustments. The authors provide extensive experimental results on NLU (GLUE), mathematical reasoning (GSM8K), code generation (HumanEval), and summarization (XSum) tasks, demonstrating that PE-DyRA consistently outperforms existing PEFT baselines. The paper also presents a theoretical analysis based on Pareto optimality to justify the superiority of a dynamic allocation strategy.

**Strengths:**

1. The paper provides compelling results across a diverse set of tasks (NLU, reasoning, code, summarization), models (DeBERTa, LLaMA 2/3), and budgets. The fact that PE-DyRA consistently outperforms baselines, including its direct predecessor AdaLoRA, strongly supports the authors' claims.
2. The ablation studies are thorough and effectively validate the two primary contributions. Table 5 clearly shows that the bidirectional strategy ("PE-DyRA") outperforms "Prune-only" and "Expand-only" variants. Table 6 demonstrates the measurable benefit of including the "input-based" importance component.
3. The paper is generally well-written and well-organized. The core idea is motivated clearly in the introduction, and the overall framework is well-illustrated in Figure 2.

**Weaknesses:**

The paper suffers from a lack of clarity in key areas, which would make reproduction extremely difficult.

1.  The hyperparameter $\alpha$ for balancing the gradient- and input-based scores ($S_{G_{i}} = \alpha \cdot S_{G_{i}}^{grad} + (1 - \alpha) \cdot S_{G_{i}}^{inp}$) is introduced in Algorithm 1 but its value is never specified anywhere in the paper.
2.  The description of the $S^{inp}$ calculation in Section 3.2.2 is ambiguous, using inconsistent indexing ($S_{ij} = |W_{ij}| \cdot \|X_j\|_2$ vs. $S_{ki}$ in the final formula).

Meanwhile, the paper's novelty is somewhat limited by its heavy reliance on AdaLoRA (Zhang et al., 2023b). The core SVD-based parameterization ($\Delta=AEB$) and the entire formulation for the gradient-based importance score (smoothed sensitivity multiplied by uncertainty) are inherited directly from this prior work. While the work is cited, the text could be clearer about the extent of this inheritance.

**Questions:**

1. The paper's core mechanism is asymmetric: pruning is fine-grained (removing the worst $k$ ranks globally), while expansion is coarse-grained (adding $k$ ranks to the top $k$ layers based on average importance).What is the rationale for this asymmetric design? Why not expand based on rank-level metrics as well? Does this design not create a "rank-churn" problem, where a new rank added to a high-average-importance layer may itself be unimportant and quickly become a candidate for pruning in the next cycle?

2. Algorithm 1 introduces a critical hyperparameter $\alpha$ to balance the gradient-based and input-based importance scores: $S_{G_{i}} = \alpha \cdot S_{G_{i}}^{grad} + (1 - \alpha) \cdot S_{G_{i}}^{inp}$. The paper only states this is "appropriately chosen". This is insufficient for reproducibility. What value of $\alpha$ was used for the experiments? Was this value fixed across all tasks and models, or was it tuned? Please provide an ablation study on the sensitivity of $\alpha$, as it seems central to the paper's contribution of combining these two importance metrics.

---

### Official Review · Reviewer_ke7J · 2025-10-30

**Soundness:** 2
**Presentation:** 3
**Contribution:** 2
**Rating:** 4
**Confidence:** 5

**Summary:**

This paper introduces a dynamic LoRA framework that adjusts rank allocation during training to improve parameter efficiency. It first computes an importance score for each LoRA component using the product of weight and gradient magnitudes, then refines this score by multiplying it with the corresponding input activation to better capture data-dependent importance insipred by model pruning.

Based on these scores, PE-DyRA performs a bidirectional rank adjustment (prune-then-extend): pruning the least important k ranks and reallocating them to the most important layers, enabling adaptive capacity redistribution under a fixed parameter budget. Experiments on GLUE, GSM8K, MATH, HumanEval, and XSum show some improvements over previous dynamic LoRA variants such as AdaLoRA and IncreLoRA, demonstrating higher efficiency and better overall performance.

**Strengths:**

1. The paper introduces a more comprehensive importance evaluation by combining input activations with gradient-based scores, leading to a more accurate assessment of parameter significance.
2. It proposes a bidirectional dynamic rank adjustment strategy that prunes less important ranks and reallocates them to critical layers, effectively improving parameter utilization and adaptability.

**Weaknesses:**

1. The performance gains diminish as the rank increases, as shown in Table 13, indicating limited scalability of the proposed method.

2. The contributions are largely incremental, as most key ideas, such as gradient-based importance scoring, activation-aware weighting, and dynamic rank allocation, have been explored in prior works. The novelty mainly lies in combining these existing elements rather than introducing a fundamentally new mechanism.

3. The improvements are relatively small on benchmarks like GLUE and XSum, and the bidirectional reallocation offers only marginal benefits compared with one-sided rank adjustment methods (Table 5).

**Questions:**

1. How sensitive is the performance of PE-DyRA to the choice of the exponential moving average (EMA) coefficients in importance estimation?
2. How is the update interval for rank adjustment chosen, and how does it affect convergence stability during training?
3. How does PE-DyRA perform compared to baselines under extreme low-rank settings(e.g., $r=1/2$)

---

### Note · Authors · 2025-12-03

I have read and agree with the venue's withdrawal policy on behalf of myself and my co-authors.